# Advances in Structural and Functional Retinal Imaging and Biomarkers for Early Detection of Diabetic Retinopathy

**DOI:** 10.3390/biomedicines12071405

**Published:** 2024-06-25

**Authors:** Zhengwei Zhang, Callie Deng, Yannis M. Paulus

**Affiliations:** 1Department of Ophthalmology, Jiangnan University Medical Center, Wuxi 214002, China; zhengwz@med.umich.edu; 2Department of Ophthalmology, Wuxi No.2 People’s Hospital, Wuxi Clinical College, Nantong University, Wuxi 214002, China; 3Department of Ophthalmology and Visual Sciences, University of Michigan, Ann Arbor, MI 48105, USA; calliede@med.umich.edu; 4Department of Biomedical Engineering, University of Michigan, 1000 Wall Street, Ann Arbor, MI 48105, USA

**Keywords:** diabetes mellitus, early diabetic retinopathy, diagnostic tests, optical coherence tomography, optical coherence tomography angiography, combined measures, deep learning

## Abstract

Diabetic retinopathy (DR), a vision-threatening microvascular complication of diabetes mellitus (DM), is a leading cause of blindness worldwide that requires early detection and intervention. However, diagnosing DR early remains challenging due to the subtle nature of initial pathological changes. This review explores developments in multimodal imaging and functional tests for early DR detection. Where conventional color fundus photography is limited in the field of view and resolution, advanced quantitative analysis of retinal vessel traits such as retinal microvascular caliber, tortuosity, and fractal dimension (FD) can provide additional prognostic value. Optical coherence tomography (OCT) has also emerged as a reliable structural imaging tool for assessing retinal and choroidal neurodegenerative changes, which show potential as early DR biomarkers. Optical coherence tomography angiography (OCTA) enables the evaluation of vascular perfusion and the contours of the foveal avascular zone (FAZ), providing valuable insights into early retinal and choroidal vascular changes. Functional tests, including multifocal electroretinography (mfERG), visual evoked potential (VEP), multifocal pupillographic objective perimetry (mfPOP), microperimetry, and contrast sensitivity (CS), offer complementary data on early functional deficits in DR. More importantly, combining structural and functional imaging data may facilitate earlier detection of DR and targeted management strategies based on disease progression. Artificial intelligence (AI) techniques show promise for automated lesion detection, risk stratification, and biomarker discovery from various imaging data. Additionally, hematological parameters, such as neutrophil–lymphocyte ratio (NLR) and neutrophil extracellular traps (NETs), may be useful in predicting DR risk and progression. Although current methods can detect early DR, there is still a need for further research and development of reliable, cost-effective methods for large-scale screening and monitoring of individuals with DM.

## 1. Introduction

Diabetic retinopathy (DR) is a leading cause of preventable blindness globally, affecting approximately one-third of patients with diabetes [1,2], with vision-related quality of life (VRQoL) declining with the severity of DR [3]. A microvascular complication of both type 1 and type 2 diabetes, DR is characterized by progressive retinal vascular abnormalities and neurodegeneration [4] and disproportionately affects individuals in socioeconomically disadvantaged areas [5]. Multiple factors contribute to the onset of DR, including chronic hyperglycemia, oxidative stress, inflammation, the complement system, and gut microbiome. These processes lead to the development of various retinal lesions, such as microaneurysms, hemorrhages, hard exudates, cotton wool spots, and neovascularization. However, the precise mechanistic pathway remains an area of active investigation [6,7,8,9].

DR can progress through distinct stages, ranging from mild non-proliferative DR (NPDR) to more advanced proliferative DR (PDR), characterized by the abnormal growth of new blood vessels on the retinal surface or into the vitreous cavity. Diabetic macular edema (DME), a consequence of increased vascular permeability and fluid accumulation in the macula, can occur at any stage and is a leading cause of vision loss among individuals with DR [10]. The impact of DR extends beyond vision impairment, affecting individuals’ quality of life, productivity, and overall well-being. Studies have shown that DR is associated with an increased risk of cardiovascular diseases, cognitive decline, and premature mortality [11,12].

Early detection and timely intervention are crucial in preventing the progression of DR and reducing the risk of vision loss and systemic cascading complications [13]. However, the initial stages of DR can be asymptomatic, making regular screening and comprehensive eye examinations essential for individuals not only diagnosed with diabetes mellitus (DM) but also those with euglycemic hyperinsulinemia or prediabetes/hyperglycemia [14,15,16]. A recent study, utilizing histological analysis and high-resolution in vivo imaging, demonstrated that subclinical DR is characterized by numerous alterations. These include changes in blood flow, vascular architecture, expression of contractile proteins, the function of pericytes and endothelial cells, glial activity and density (including astrocytes, Müller cells, and microglia), neuronal function, retinal cell counts, layer thickness, and choroidal thickness [17]. These collective subclinical changes likely contribute to the ultimate manifestation of clinically evident DR. The first sign of clinical DR—formation of microaneurysms—may represent a relatively late stage in the disease process. It is reasonable to deduce that the onset of clinically detectable DR lesions is preceded by a cascade of subclinical pathological events, underscoring the importance of identifying and monitoring these early alterations for timely intervention.

The significance of early detection of DR is twofold. From the ophthalmologist’s perspective, advanced imaging modalities allow for the identification of subtle retinal and choroidal alterations at the earliest stages of the disease process. This knowledge guides the development of therapeutic interventions targeting these initial pathological events, enabling clinicians to promptly intervene and advise patients on stricter management of controllable risk factors. From the patient’s perspective, convenient screening enables regular examination and early detection of DR, empowering patients to take a proactive role in managing their disease and preserving vision [18].

Advanced imaging and functional testing, combined with accurate risk stratification and personalized treatment planning, can improve visual outcomes for individuals with diabetes. This review synthesizes the latest advancements in multimodal imaging and functional testing for DR detection while highlighting the value of monitoring subclinical retinal alterations for early intervention.

The search strategy is as follows: Medline, Embase, and Google Scholar were searched until 1 April 2024. The keywords used in the search were “diabetic retinopathy”, “prediabetes/hyperglycemia”, and “early detection”. The reference lists of relevant reviews and eligible articles identified from the electronic searches were manually examined to locate any pertinent studies that were not included in the electronic databases. Furthermore, the related articles cited in Google Scholar were also manually reviewed to identify any potentially relevant studies that had not been retrieved by the initial search. 

## 2. Fundus Photography and Image Analysis

The retinal vasculature presents a uniquely accessible “window” for studying human microcirculation in vivo. Advancements in digital retinal photography and other imaging techniques have facilitated the precise characterization of retinal vascular changes in large populations. This non-invasive imaging approach has enabled researchers to investigate early retinal microvascular alterations as potential biomarkers for systemic microvascular diseases [19,20,21].

### 2.1. Conventional Color Fundus Photography

Conventional color fundus photography, capturing a field of view between 20° and 50°, has long been the standard method for screening and evaluating DR. This technique involves capturing images of the posterior retina using a tabletop fundus camera. In recent years, the emergence of portable and user-friendly imaging devices has made it easier to perform retinal screenings, particularly in remote or resource-limited settings [22,23]. The widespread availability of smartphones with advanced camera capabilities has led to the development of mobile phone-based fundus imaging systems (Figure 1) [24,25,26,27,28]. These systems typically involve attaching a lens or adaptor to a smartphone camera and transforming it into a portable fundus camera. This approach leverages the computational power and connectivity of smartphones, enabling telemedicine applications where retinal images can be captured and transmitted for remote evaluation by ophthalmologists or artificial intelligence (AI) systems [29,30].

Up until now, color fundus photography remains a simple, accessible, and cost-effective technique for examining the retinal fundus. However, it is important to note that while the appearance of microaneurysms on color fundus photographs represents an early clinical manifestation, advanced multimodal imaging techniques and functional tests have revealed subclinical retinal alterations that occur long before the manifestation of these minimal fundus lesions [13,32,33]. A comparison of retinal imaging modalities, including their advantages, disadvantages, and key findings associated with early DR, are summarized (Table 1) and explored in further detail below. It is crucial for ophthalmologists to acknowledge that even in the absence of visible lesions in fundus photography, special attention is needed for high-risk diabetic patients with a disease duration exceeding 10 years. Alongside advising patients to undergo regular screening for potential complications, it is advisable to suggest auxiliary examinations for comprehensive evaluation and monitoring [34]. 

### 2.2. Ultra-Wide-Field Fundus Photography

Recent advancements in fundus imaging enable physicians to capture a comprehensive view of the retina, covering up to 200° in a single image [35]. Clarus™ (CLARUS 500™, Carl Zeiss Meditec AG, Jena, Germany) and Optos^®^ (Optos California^®^, Optos PLC, Dunfermline, UK) are currently the two most used ultra-wide-field (UWF) fundus imaging systems. Although the reliability in detecting signs of early DR is high for both devices [36], the Clarus™ system has demonstrated higher accuracy in detecting microaneurysms and retinal hemorrhages due to its capability of providing true-color retinal images and mitigating artifacts caused by eyelids and eyelashes [37]. Diabetic eyes exhibiting visible lesions exclusively in the periphery tend to present with a milder and earlier form of retinopathy in comparison to those with visible lesions present in either the central retina or spanning both the central and peripheral retina [38]. Therefore, comprehensive visualization and assessment of the peripheral retina are crucial for early detection, accurate severity staging, and prediction of DR progression [39,40,41]. In clinical practice, pupillary dilation and manual eyelid lifting have been reported to substantially increase the visible retinal area during UWF fundus imaging, enhancing the detection of retinal hemorrhages and microaneurysms [42]. To accurately detect early, subtle lesions, operators must pay close attention to the technique used when capturing UWF fundus images.

### 2.3. Advanced Fundus Image Analysis Techniques

#### 2.3.1. Retinal Vascular Caliber

Pathophysiologic processes central to diabetes such as oxidative stress, endothelial dysfunction, inflammation, and hypertension can cause subclinical alterations in microvascular structure and perfusion. Abnormalities in retinal vessel caliber such as venular widening and microaneurysms can, therefore, serve as promising biomarkers for diabetic microvascular complications [43].

Wider retinal venular caliber has been correlated with an elevated risk of type 2 diabetes. In an extensive individual-level meta-analysis comprising 18,771 participants, retinal venular caliber was positively associated with an increased risk of developing diabetes over a median follow-up period of 10 years [21]. This correlation remained significant even after adjusting for potential confounders, including age, race/ethnicity, smoking status, body mass index (BMI), and hypertension, and was notably more pronounced in male participants. There was no significant correlation between retinal arteriolar caliber and incident type 2 diabetes [21]. In addition to predicting the risk of developing diabetes, studies suggest that venular caliber is significantly wider among type 2 diabetics with DR compared to non-DR [44]. However, the underlying mechanisms driving changes in retinal venular caliber in diabetes are not fully understood. Experimental data suggest that insulin resistance-induced microvascular dysfunction may prompt retinal venule dilation due to inflammation [44,45]. Monitoring changes in retinal vascular caliber could aid in the early detection of DR.

#### 2.3.2. Tortuosity of Branch Retinal Artery

High glucose levels similarly precipitate changes in retinal vessel tortuosity. Increased vessel tortuosity, prevalent in various diseases and aging, serves as a critical indicator of retinal ischemia and DR progression [46]. Tortuosity reflects diabetes-induced hemodynamic alterations, including disrupted blood flow and endothelial dysfunction, alongside elevated vascular endothelial growth factor (VEGF) production.

Studies on the associations between retinal arteriolar tortuosity, venular tortuosity, and DR are mixed. In a study of 224 type 1 and 2 diabetic patients, Sasongko et al. reported that increased arteriolar tortuosity, not venular tortuosity, is associated with mild and moderate stages of DR [46]. Conversely, Forster et al. reported that greater venular tortuosity alone is associated with incident DR in adults with type 2 diabetes [47]. A recent study developed a prediction model for DR among type 2 diabetics and found that while arteriolar and venular tortuosity are both associated with DR, arteriolar tortuosity is significantly more predictive of DR [48].

Fundus photography-based studies investigating vessel tortuosity have primarily concentrated on the main retinal vessels, partly due to challenges in visualizing and analyzing branch retinal vessels in fundus images. A recent study involving high-resolution fundus images from diabetic patients across varying DR stages proposed that branch retinal vessels may be more closely related to the onset and progression of DR than main vessels [49]. Findings revealed a significant increase in branch artery tortuosity from no DR to evident DR and with increasing DR severity. Thus, branch retinal artery tortuosity appears to be a promising biomarker for early DR detection and precise evaluation, aiding in effective DR management strategies.

#### 2.3.3. Fractal Dimension

Fractal dimension (FD) quantifies the complexity or density of vascular patterns in a two-dimensional space. It is more closely linked to microvascular than macrovascular diseases [48]. In DR, disease progression is marked by the formation of non-perfusion areas due to occluded microvasculature. This often leads to a simplified retinal vascular pattern and a decreased FD. However, some studies have reported an increase in FD in DR, highlighting the complexity of vascular changes throughout the disease’s progression [50]. From moderate non-proliferative to proliferative stages characterized by neovascularization, FD may vary significantly. Observational studies in younger individuals with type 1 diabetes have shown an association between lower FD and proliferative DR [51,52]. In a prospective study of adults with type 2 diabetes, decreased FD at baseline was independently associated with the development of DR over a 10-year follow-up, even when accounting for other risk factors [47]. This was corroborated by a recent retrospective study, which identified decreased FD as one of five predictive variables for DR among type 2 diabetic patients, suggesting a potential role in early detection and assessment of DR [48].

## 3. Structural Imaging: OCT and OCTA

### 3.1. Structural Changes in Early DR

#### 3.1.1. Retinal Neurodegeneration

The ganglion cell–inner plexiform layer (GC-IPL) is a vital neural layer in the retina. Changes detected in this layer can be indicative of neurodegeneration and can potentially occur in patients with DM even before the development of DR. This may be due to damage from chronically elevated blood sugar levels [53,54,55]. A 3-year longitudinal study confirmed significantly faster rates of GC-IPL thinning in eyes that developed incident DR compared to those that remained non-DR, although both groups showed decreased thickness [56].

Apart from the GC-IPL, the peripapillary retinal nerve fiber layer (pRNFL) and macular retinal nerve fiber layer (mRNFL) are other important retinal structural parameters that need to be considered for early detection of DR [57,58,59,60]. More recently, Hafner et al. reported a significant association between pRNFL but not mRNFL with parafoveal vessel density detected by OCTA [55]. This finding corroborated the result reported by the EUROCONDOR study, which demonstrated a strong correlation between narrower retinal arteriolar caliber and thinning of the pRNFL, reflecting the close relationship between microvascular abnormalities and neurodegeneration in the pathophysiology of diabetic retinopathy [61]. Therefore, monitoring GC-IPL and pRNFL thickness changes may offer a valuable window for early detection of DR.

#### 3.1.2. Retinal Reflectivity

Structural OCT allows for quantitative assessment of retinal reflectivity, which holds potential in early diagnosis and prognostication of various retinal diseases, including DR [62,63]. Retinal reflectivity is a measure of the intensity of backscattered light from retinal tissues that can provide insights into structural and compositional changes associated with neurodegenerative processes and vascular impairment in early DR. Zhang et al. reported that outer retinal reflectivity was significantly reduced in non-DR diabetic patients when compared with normal controls, especially for the ellipsoid zone (EZ) [64]. Concurrently, other researchers investigated the role of inner retinal reflectivity. In a study conducted by Cetin et al. [65], diabetic patients without clinically evident DR exhibited a significant correlation between ganglion cell layer reflectivity and the extent of pericentral retinal thinning over time. Notably, this retinal thinning was more pronounced in the inner retinal layers, which comprise the ganglion cell and inner plexiform layers. However, a notable limitation of retinal reflectivity is the requirement for post-acquisition processing and analysis using third-party software (i.e., ImageJ, https://imagej.net/ij/). OCT images must be exported and imported into specialized software for quantitative analysis of reflectivity patterns, which can be time-consuming and impractical in clinical settings. If this functionality is integrated into commercial OCT devices in the future, it could streamline clinical applications.

#### 3.1.3. Choroidal Vessel Index and Choroidal Thickness

Studies indicate that choroidal vascular index (CVI) and choroidal thickness (ChT) can be used to quantitatively assess changes in the structure and blood flow of the choroid, potentially serving as early biomarkers and severity indicators of DR [66,67,68]. Three studies with ultra-wide-field swept-source OCT (SS-OCT) revealed that diabetic patients exhibited significantly lower values of CVI and ChT compared to healthy individuals, with the difference being more pronounced in patients with early-stage DR than in those without clinical DR. Of note, the peripheral choroidal capillaries are more susceptible to the early microvascular insults induced by DM than the central choroidal region [69,70,71]. The observed reduction in choroidal vascularity and thinning of the choroid may precede clinically detectable retinal vascular changes.

Eyes that develop incident DR show significantly faster rates of ChT thinning compared to those that remain non-DR [56]. Remarkably, the peripheral choroidal alterations appear more pronounced than changes in the posterior pole during the early stages of DR. This discrepancy may represent a compensatory mechanism, wherein the choroid attempts to maintain adequate blood supply to the metabolically demanding posterior pole retina by redistributing flow from the periphery. These findings underscore the importance of closely monitoring peripheral choroidal changes, as they may serve as early indicators of the deleterious effects of diabetes on the choroidal–retinal complex.

### 3.2. Vascular Changes in Early DR

In contrast to fundus fluorescein angiography (FFA), which necessitates intravenous injection of a contrast agent, OCTA is a non-invasive, dye-free, three-dimensional imaging technique [72]. It not only allows visualization of capillaries across all retinal layers but also provides a broader perspective of the fundus, facilitating comprehensive examination and quantitative analysis [73]. Researchers have conducted numerous studies validating OCTA’s ability to detect early signs of microvascular alterations. Additionally, it provides a quantitative assessment of disease severity, including conditions like DR and its associated complications [74,75,76]. Recently, a meta-analysis suggests retinal microvascular damage may precede clinical DR and can be detected early using OCTA [33].

#### 3.2.1. Retinal and Choroidal Vascular Density and Perfusion

A study focused on retinal perfusion reported that the perfusion density of the deep capillary complex (DCC) was notably reduced in patients with diabetes, even in the absence of clinically detectable DR. Remarkably, the retinal capillary network outside the parafoveal region exhibited greater susceptibility to capillary perfusion deficits compared to the capillaries close to the fovea [77]. Investigating images of 12 mm × 12 mm scanning area obtained by swept-source OCTA (SS-OCTA), Qi and colleagues’ study unveiled that structural and blood flow alterations in the choroid manifested before the onset of DR and preceded changes in the retinal microcirculation [78]. Of note, their findings highlighted that mid-large choroidal vessel thickness and density were more sensitive imaging biomarkers for the early clinical detection of DR compared to retinal vascular changes. Interestingly, the choriocapillaris remained unaffected in the pre-clinical and early stages of DR [78], possibly due to the necessity for compensatory blood supply. This finding was corroborated by results from another study [79]. These results suggest that parameters of choroidal blood vessels could serve as earlier indicators of diabetic fundus changes, potentially enabling more timely detection and intervention before retinal vascular complications arise.

While the aforementioned studies highlight that the choriocapillaris remains unaffected in the pre-clinical and early stages of DR, choriocapillaris perfusion metrics (quantitative flow deficit density, number, and size) in the macula from another study were reported to independently correspond to the severity of DR [80]. This suggests that although the choriocapillaris may not exhibit observable changes in the initial stages, its perfusion characteristics could serve as valuable biomarkers for evaluating the progression and severity of DR. Moreover, two studies reported a significant decrease in choriocapillaris perfusion of the central macular region (3 mm × 3 mm or 6 mm × 6 mm) in diabetic patients without DR when compared to age-matched non-diabetic controls. This reduction in choriocapillaris perfusion was observed despite an absence of detectable changes in macular retinal vessel parameters [81,82]. These findings suggest that decreased choriocapillaris perfusion in the macular region may serve as an early indicator of diabetic vasculopathy, potentially preceding clinically apparent retinal vascular alterations.

#### 3.2.2. Ultra-Wide-Field SS-OCTA for Vascular Analysis

The TowardPi SS-OCT/OCTA commercial system (Medical Technology, Beijing, China) is a cutting-edge, high-resolution, and wide-field tomographic and angiography imaging platform [83]. Boasting an impressive A-scan rate of 400 KHz and an axial scan depth of 6 mm, this advanced system offers unprecedented capabilities. Remarkably, it can acquire wide-field tomography spanning 24 mm and comprehensive angiography information covering an extensive 24 × 20 mm area in a single rapid capture, taking only about 15 s (Figure 2). This innovative system combines high-speed imaging with an extensive field of view, enabling comprehensive and efficient examination of retinal and choroidal structures [84,85].

An observational study demonstrated that the average vascular density (VD) of the superficial vascular complex (SVC) across all observed areas was significantly lower in DR eyes compared to normal controls [86]. However, only the most peripheral area (16–21 mm) exhibited a significant decrease in VD for the DM group relative to healthy individuals, with a good receiver operating characteristic (ROC) curve value (0.8353) to predict DR. The average VD of the deep vascular complex (DVC) in the outermost peripheral area (16–21 mm) was significantly reduced in the DM group compared to the normal controls. Surprisingly, no significant changes were observed in the thicknesses of the SVC or DVC nourishing segments in the peripheral retina. These findings suggest that alterations in peripheral choroidal blood flow, as reflected by vascular density changes, may serve as a more sensitive indicator for early detection of diabetic fundus changes than thickness measurements alone.

#### 3.2.3. Foveal Avascular Zone

The foveal avascular zone (FAZ) is a specialized, capillary-free region at the center of the macula, which is essential for preserving high visual acuity. It appears as a circular or slightly oval-shaped area devoid of retinal vasculature. OCTA enables detailed, non-invasive visualization and quantitative analysis of the FAZ’s size, perimeter, shape, and vascular density, facilitating early detection and monitoring of macular disorders affecting this critical area [74,87]. Although FAZ size can exhibit great inter-person variability among normal individuals (Figure 3) [88,89], longitudinal follow-up and monitoring of FAZ dimensions through OCTA may prove valuable in detecting and tracking the progression of DR [90,91]. By contrast, the FAZ circularity index serves as a relatively stable parameter that reflects the regularity of the FAZ shape [92,93], with a mean value of 1.12 to 1.32 in normal subjects [93,94]. With the progression of DR severity, the FAZ is recognized to exhibit increased tortuosity or irregularity in its shape [94]. This alteration in FAZ contour is attributed to localized capillary dropout at various points along its perimeter, resulting in an expanded and less circular outline. The change in the FAZ circularity index may serve as a valuable biomarker for monitoring the progression and severity of DR, especially in instances where longitudinal data are unavailable.

## 4. Functional Tests

### 4.1. Multifocal Electroretinogram (mERG)

Multifocal electroretinography (mfERG) has been widely used to evaluate the retinal function of the macula. DR is largely caused by defects of the retinal deep capillary plexus in the inner nuclear layer, where the cell bodies of bipolar cells are located. It has been documented that the primary generators of mfERG are on and off bipolar cells [95]. Thus, mfERG is well suited for the study of DR. By precisely mapping localized retinal responses, mfERG can detect early neurodegenerative alterations associated with DR, even before clinically evident vascular lesions appear. By utilizing electroretinography (ERG), some studies have demonstrated that retinal dysfunction can be detected in children with DM without visual impairment or clinically evident DR [96,97]. Their findings suggest that hyperglycemia can induce neurodegenerative changes in the retina during the early stages of type 1 DM, even before the onset of overt vascular lesions or visual symptoms. These results highlight the potential of ERG or mfERG as sensitive functional tests for detecting subclinical retinal neurodegeneration, which may precede the microvascular complications of DR.

Recent mfERG studies have found that both reduced amplitudes and response delays are present in DM patients with or without clinical DR. Additionally, the degree of mfERG implicit time delays appears to be directly related to the severity of DR, and the locations of abnormal implicit times spatially align with anatomical abnormalities in the macula [98]. The areas of the retina that appeared visually normal in clinical examinations but exhibited functional abnormalities showed overall stability. However, after a one-year follow-up, these areas were found to be more susceptible to the development of microaneurysms compared to zones with a normal baseline implicit time [99]. In short, mfERG can be a valuable monitoring tool and effectively identify early abnormalities of the retina in DM patients without or with early stages of DR [100,101,102]. Notably, a mfERG study suggested that males may be more vulnerable to the neurodegenerative changes that preceded the development of background diabetic retinopathy in type 2 diabetes, compared to their female counterparts [103].

### 4.2. Microperimetry

Microperimetry (MP) is a visual function test that maps light sensitivity across the retina. Participants actively report their perception of light stimuli presented at various locations, generating a detailed decibel (dB) map of retinal function. Unlike other functional tests like electroretinography and standard perimetry, MP’s ability to register the fundus allows for correlations between functional changes (light sensitivity) and the underlying retinal structure [104]. MP-3, the state-of-the-art model of fundus microperimetry, boasts a wider range of measurable parameters and eye tracking system. Early detection of DR can benefit from measuring retinal sensitivity, as documented in many studies [105]. It can reveal subtle declines in retinal sensitivity, even before structural changes are visible through ophthalmoscopy or OCT/OCTA. While the full implications of this finding are still being debated, microperimetry appears valuable for identifying early functional impairment in the diabetic retina. This technique excels at mapping the point-to-point relationship between retinal structure and function.

### 4.3. Multifocal Pupillographic Objective Perimetry

Multifocal pupillographic objective perimetry (mfPOP) is a rapid and objective visual field test that measures how pupils react to multiple light stimuli flashed simultaneously. This response can help track retinal dysfunction in diabetic patients, potentially reflecting the severity of underlying blood vessel damage in the retina [32,106,107]. Compared with subjective automated perimetry (SAP), Sabeti et al. found significant differences between the non-proliferative diabetic retinopathy for ObjectiveFIELD Analyzer (OFA; Konan Medical USA, Irvine, CA) mean defects (MDs) and pattern standard deviations (PSDs), but not for SAP MDs or PSDs [108].

### 4.4. Contrast Sensitivity

Contrast sensitivity (CS) goes beyond just detecting light intensity differences, providing a more comprehensive assessment of spatial vision. Research has shown it to be associated with overall health function [109]. Recently, Silva-Viguera et al. conducted a systematic review including a total of 21 studies published between 2010 and 2021 to evaluate whether CS assessment in patients with type 1 or type 2 diabetes could be a reliable test in early detection of DR [110]. The study concluded that individuals with DR exhibited a substantial reduction in CS across a wide range of spatial frequencies. The findings have been supported by recent research publications, which echo similar conclusions [111,112,113]. While some diabetic patients without clinically evident DR may also demonstrate decreased CS, this effect was less consistent, and the affected frequencies varied. These changes in visual function suggest that retinal neuronal damage may precede overt vascular lesions in DR, potentially enabling early detection through CS testing. However, longitudinal studies are warranted to establish CS as a reliable biomarker for predicting DR onset and progression, strengthening evidence for its application in early screening and monitoring [112].

### 4.5. Visual Evoked Potential Test

The visual evoked potential (VEP) test offers a functional assessment of the integrity of the optic nerve. Using the pattern-reversal VEP (PRVEP) test, the technique showing less variability in timing and waveform, El-Tawab et al. found a significant delay in P100 latencies of the patients with pre-clinical DR when compared with normal controls [114]. The findings indicated retinal ganglion cell dysfunction and demyelinating changes in the optic nerve pathway, which are consequences of microvascular insults caused by the hyperglycemic state in diabetes [115]. Although the VEP test is not routinely used for DR, it can be considered for early detection of DR in patients who are negative in other tests but need to rule out whether there is retinal dysfunction.

### 4.6. Retinal Vessel Reactivity

In current clinical practice, two stimuli prompt responses from retinal blood vessels, facilitating the assessment of vascular function: flicker light and gas perturbation. The Dynamic Vessel Analyzer (DVA, Imedos, Jena, Germany) is a commercially available system comprising a fundus camera and a video unit connected to a computer. This setup facilitates uninterrupted recording of the retina throughout the examination, with specialized software monitoring changes in retinal vessel diameter in real-time in response to diffuse luminance flicker [116]. It has been reported that prediabetic and diabetic individuals without DR exhibit significantly attenuated peak vasodilator responses and relative amplitude changes in retinal vein and artery diameters in response to the flickering light stimulus compared to healthy controls [117]. Further studies have demonstrated that the responses of retinal arterioles and venules to flickering light are diminished in subjects with DR, and this reduction progresses with more severe stages of DR [118]. Another prospective study unveiled that decreased dilatory responses of retinal arterioles and venules to flickering light are linked to a greater probability of DR progression within one year among adult patients with DM [119]. These findings may indicate endothelial dysfunction or disrupted neuro-regulation of retinal vascular tone in eyes affected by DM or DR, despite limited reports of negative results [120].

The other approach for evaluating retinal vascular responses is gas perturbation experiments that involve modifying the partial pressures of CO_2_ and O_2_ (PCO_2_ and PO_2_) within the bloodstream. The alterations in retinal microvascular blood flow signals can be captured by OCTA. Safi et al. assessed the retinal vascular response to hyperoxia in patients with diabetes at the pre-clinical stage of DR and compared it with normal controls. They found that impaired retinal vascular reactivity was apparent at this stage, with disturbances in the autoregulatory mechanism particularly pronounced in the parafoveal DCP [121]. This finding further underscored that the DCP experiences more severe microvascular damage than the SCP in patients with DM or DR at the early stage [122]. In another study on patients with mild to severe DR, findings revealed impaired retinal capillary reactivity across the full retinal layer to both hyperoxia and hypercapnia among individuals with DM in comparison to healthy controls [123]. However, the clinical practicality of this method is limited, and it may serve as a supplementary approach.

## 5. Combining Structural and Functional Data

Vascular and neurodegenerative changes occur early in the DR disease process, causing loss of function and cell death among retinal ganglion cells, with consequent GC-IPL and RNFL thinning. Numerous studies have looked at the integration of multimodal imaging data to elucidate the pathophysiologic relationship between retinal neurodegenerative and vascular changes prior to the appearance of DR. Emerging evidence points to early neuronal damage as a precursor to clinically visible retinopathy changes. Combining structural and functional imaging data may facilitate earlier detection of DR and targeted management strategies based on disease progression.

In a recent study, Boned-Murillo et al. examined patient eyes with type 2 diabetes using SS-OCT and microperimetry, correlating macular Early Treatment Diabetic Retinopathy Study (ETDRS) grid areas with corresponding microperimetry points [124]. Total retinal thinning was found at the parafoveal ring, with a reduction in both ganglion cell layer (GCL) and inner plexiform layer (IPL) thickness, suggesting ganglion cells have increased susceptibility to neurodegenerative and vascular effects in DM patients. Additionally, both mRNFL thickness and retinal sensitivity were decreased in patients with moderate DR with no DME. In particular, the RNFL thickness was significantly lower in the outer nasal area in ganglion cells and IPL among T2DM patients. While retinal sensitivity was correlated with RNFL thickness, there was no correlation between retinal sensitivity and thickness of the GCL combined with the IPL.

Topographic quantification of macular function using microperimetry has also shown utility in assessing mild DR, with studies successfully detecting functional impairment [125]. In mild DR eyes, studies suggest that mesopic macular dysfunction may be present with a preserved outer retinal thickness and a strong relationship between macular perfusion and photoreceptor function.

Several studies have also examined the relationship between OCTA-measured blood flow channels and functional parameters [126,127,128]. In a prospective observational study, Tsai et al. applied OCTA and microperimetry to examine the relationship between perifoveal vessel densities, FAZ areas, and retinal sensitivity in diabetic patients [126]. They found deep perifoveal vessel density was inversely correlated with the severity of DR and directly correlated with retinal sensitivity. Meanwhile, the superficial FAZ area was inversely correlated with retinal sensitivity. Further building on this, Levine et al. registered OCTA images with microperimetry of eyes stratified by severity of DR to quantitatively demonstrate a point-wise structure–function relationship between vessel density obtained from OCTA and retinal sensitivity in global and zonal spatial scales in the parafovea [128]. Notably, the study identified local regions of retinal flow impairments that were only evident in advanced DR and were most often accompanied by regions of functional loss, but not vice versa.

This suggests that local ischemia may not be the initial cause of retinal dysfunction in DR, but rather supports the neurodegenerative theory of DR. The neurodegenerative theory posits that DR primarily affects retinal neurons and the reaction of these retinal neurons to microenvironment stressors is responsible for vascular complications. This operates in contrast with the classic hypothesis of DR as primarily a retinal vascular disorder with pathogenesis rooted in glycosylation-induced microvascular damage that leads to ischemic neuronal injury. Significant research has been published suggesting both neurodegeneration and glial dysfunction precede microvascular changes in DR. Studies using mfERG to assess neuroglial dysfunction have shown an increase in implicit time is predictive for the development of visible vascular abnormalities over 1-year and 3-year periods [129,130].

Alterations in neuronal function measured by mfERG similarly precede clinical manifestation of structural changes in the ganglion cell complex and retinal vascular changes in prediabetics. Studies show a significant decrease in mfERG amplitude but no significant increase in implicit time among prediabetics [131,132]. This is partially corroborated by the EUROCONDOR study where mfERG P1 amplitude was seen to be more sensitive than the P1 implicit time [133]. Combining OCTA with mfERG data, Zagst et al. showed a significant positive correlation between FAZ area and mfERG amplitude [132].

Another study by Srinivasan et al. aimed to determine structural and functional differences between eyes with and without NPDR using OCTA and mfERG [100]. The group with mild or moderate NPDR had no statistically significant structural differences from those without DR. However, participants with mild to moderate NDPR had significantly lower mfERG implicit times and response densities in rings five and six, suggestive of abnormal neuronal function. In eyes without DR, lower macular vessel density and perfusion almost always corresponded to delayed implicit times and lower response densities. However, these correlations were not significant in the NDPR group, suggesting that other structural and functional correlates could exist in retinas with NPDR. These findings were expanded upon in a study by Santos et al. that more granularly stratified the NDR group and compared them to healthy controls [133]. DR was correlated with higher implicit times of mfERG rings 3–6 and lower response density. Structurally, diabetic patients with ETDRS <20 had significantly thinner GC-IPLs compared to non-diabetics. However, in retinas with ETDRS levels between 20 and 35, there was no significant relationship between GC-IPL thickness and ETDRS level.

## 6. Applications of Artificial Intelligence in DR Detection

Artificial intelligence (AI) such as deep learning (DL) algorithms can be trained on large datasets of retinal images labeled by ophthalmologists to provide automated DR diagnosis or risk assessment. Current applications of AI demonstrate remarkable sensitivity and specificity in detecting referable DR from retinal fundus photographs [134,135] and show the potential to help efficiently triage patients, optimize resource allocation, and reduce the burden on healthcare systems. A recent meta-analysis of 21 prospective studies investigating diagnostic performance of AI algorithms for DR showed a pooled sensitivity of 88% (87.5–88.4%), pooled specificity of 91.2% (99–91.3%), AUC of 0.98, and pooled diagnostic odds ratio of 206.80 (124.82–342.63) [136]. Higher quality studies in the analysis demonstrated less heterogeneity in performance. Studies with higher image quality, greater number of included eyes, patient sourcing from healthcare facilities, and high population representativeness yielded higher diagnostic efficacy. The integration of AI-based automated tools offers substantial benefits including reduced screening costs, improved accessibility to healthcare services, and facilitation of earlier interventions and treatments [137].

### 6.1. Milestones and Market Approvals for AI Algorithms

Historical milestones for applications of AI in DR include the 2016 Google deep learning DR validation study by Gulshan et al., which presented a deep convolutional neural network (CNN) trained on a vast dataset of 128,175 fundus photos to detect referable DR, defined as moderate or worse DR or referable DME [138]. In this retrospective study, photos were acquired in the US and three eye hospitals in India using a variety of cameras. Compared to the reference standard grading determined by a panel of US board-certified ophthalmologists, the DL model demonstrated sensitivities greater than 96% and specificities greater than 93% for detection of referable diabetic retinopathy. Concomitantly, Gargeya et al. developed a customized CNN for detecting DR and validated it against a larger dataset of color fundus photos from multiple independent sources, emphasizing the generalizability of their model. In addition to showing comparable sensitivity and specificity for the detection of referable DR, Gargeya et al. demonstrated the utility of a DL algorithm to detect mild DR, with sensitivities ranging from 74–90% and specificities ranging from 80–94% [139].

In 2018, the US Food and Drug Administration (FDA) approved IDx-DR (now rebranded as LumineticsCore), the first AI-powered medical device for analyzing retinal images in primary care settings for DR screening. In a multi-center pivotal trial for IDx-DR, Abramoff et al. prospectively enrolled 900 diabetic adult participants without a prior diagnosis of DR and acquired two-field retinal fundus photos centered on the disc and macula, using a Topcon TRC-NW400 camera (Topcon, Tokyo, Japan) [140]. Compared to the reference standard of experienced human graders using the ETDRS severity scale, the IDx-DR algorithm demonstrated a sensitivity of 87% and specificity of 91% for detecting more than mild DR (mtmDR), defined as ETDRS > 35.

To date, three AI algorithms for DR screening have been approved by the FDA: LumineticsCore (formerly known as IDx-DR), EyeArt, and AEYE-DS. Since its approval in 2018, the LumineticsCore algorithm has been improved to read previously ungradable images with higher processing speeds. EyeArt, a deep learning algorithm developed by Eyenuk and first FDA-cleared in 2020, was similarly designed to screen patients for referable DR based on the ETDRS standard (>35) using two-field color fundus photos. In a US multi-center pivotal trial, EyeArt applied the same protocol implemented by authors of IDx-DR to show 96% sensitivity and 88% specificity for detecting mtmDR [141]. Most recently, AEYE-DS, an algorithm created by AEYE Health, received FDA clearance in April 2024 as the first fully autonomous AI algorithm that diagnoses referable diabetic retinopathy from retinal images taken by a handheld camera. While the details of the pivotal studies have yet to be published as scientific manuscripts, in a recent press release AEYE-DS was reported to have achieved sensitivities ranging from 92–93% and specificities between 89–94%, across two large-scale prospective phase 3 studies.

Several AI-based algorithms and devices have received class IIa approval in the European Union (EU), including LumineticsCore, EyeArt, RetmarkerDR by Retmarker, SELENA+ by eyRIS, Automated Retinal Disease Assessment (ARDA) by Verily Life Sciences LLC, Medios AI by Remidio, OphtAI by Evolucare, VUNO Med-Fundus AI by VUNO Inc., and RetCAD by Thirona (Table 2) [142,143].

### 6.2. OCT/OCTA-Trained AI Algorithms

While the majority of DR screening algorithms have been developed for color fundus photos, recent studies have explored training AI algorithms on OCT and/or OCTA images [144,145], or UWF OCTA images [146]. Compared to fundus photography, OCTA has only recently been introduced into clinical eye care over the last 10 years and the availability of large, publicly available datasets remains relatively limited. However, given DR’s nature as a retinal microvascular disease, OCTA’s detailed microvascular insights significantly enhance diagnosis, particularly through metrics like the FAZ area and macular capillary density, which correlate with DR pathophysiology. OCTA’s advantages over fluorescein angiography, such as its ability to image both deep and superficial vascular plexuses, non-invasiveness, ease of use, and compatibility with existing OCT platforms, make it an invaluable complement to standard OCT.

In a single-center, cross-sectional study based in South Korea, Ryu et al. developed a fully automated CNN-based classification algorithm that detects the onset of DR and referable status using raw OCTA images. Their model was able to detect early DR with an accuracy of 90–95%, sensitivity of 91–98%, and specificity of 85–93%, comparable to previously reported CNN models utilizing fundus photographs with fewer than 250 samples [147]. The study also explored the algorithm’s performance across various image sizes and retinal slabs to pinpoint optimal configurations for DR classification, finding the end-to-end CNN classifier surpassed traditional machine learning approaches that relied on extracted local features from OCTA images. Another study achieved robust model performance by not only including the raw OCTA images but also extracting a binary map of the segmented retinal vascular network from OCTA images and calculating a distance map of blood vessels as additional processed inputs for the CNN model [148].

Sandhu et al. developed a novel machine learning algorithm for diagnosing and grading NPDR by integrating OCT and OCTA imaging data with basic clinical and demographic information from 111 patients [149]. Three pathophysiologic features were extracted from each layer on OCT: reflectivity, curvature, and thickness; four features were extracted from each OCTA plexus: blood vessel caliber, vessel density, size of FAZ, and the number of bifurcation and crossover points. Combined with clinical data, these seven image features were fed into a two-stage random forest classifier, which first determined the presence of DR and subsequently classified the severity of NPDR with 96% accuracy, 100% sensitivity, and 94% specificity. The model outperformed machine learning models trained on OCT or OCTA images alone, demonstrating the utility of combining OCT and OCTA imaging with patient data to improve the detection and classification of NPDR. As OCT technology becomes more ubiquitous, there may be significant potential for its application in DR screening.

### 6.3. Potential Risks and Limitations

A significant risk of AI models is the potential for bias. This can come in the form of data sampling and representation bias, where non-random sampling produces datasets that do not represent the diversity of the population. As a result, a model trained on a biased dataset may not generalize to data collected from a new population, performing poorly for certain subgroups [150]. For example, an algorithm trained predominantly on retinal images of a subgroup with lighter skin pigmentation and lower retinal pigment scores may not accurately diagnose retinal disease in individuals with darker skin pigmentation and higher retinal pigment scores [151]. Additionally, there is also potential for algorithmic bias, where bias is not present in the input data and is added purely by the algorithm. Bias can exacerbate existing healthcare disparities, particularly affecting under-represented and marginalized populations. To mitigate this risk, developers must ensure that training datasets are diverse and representative of the broader population.

The quality and variability of data used to train AI algorithms also pose significant challenges. Retinal images can vary widely based on the equipment used, the skill of the operator, and the conditions under which the images are taken. Variations in image quality can impact the performance of AI systems, potentially leading to false positives or negatives. Ensuring consistent, high-quality data for training and validation is essential but challenging in diverse clinical environments. Standardizing imaging protocols and using high-quality datasets can help improve the robustness of AI systems.

### 6.4. Practical Implementation Challenges

#### 6.4.1. Integration into Clinical Workflows

Despite the proven accuracy and reliability of DL systems in detecting various ocular diseases, their integration into clinical practice faces hurdles, primarily due to concerns over the systems’ interpretability and complexity. Efforts to improve neural network visualization techniques have progressed, offering insights into decision-making aspects of AI, although inconsistencies among visualization methods exist. Healthcare providers must ensure that these technologies complement rather than disrupt current practices. This requires a careful redesign of workflows to incorporate AI screening processes without overburdening healthcare professionals or causing delays in patient care. Medical professionals also need to be trained to use new AI systems, interpret the results, and understand the limitations of AI-based diagnostics. Addressing these concerns through comprehensive training programs and demonstrating the efficacy of AI systems through pilot projects and case studies can help mitigate resistance and build trust among healthcare providers.

#### 6.4.2. Cost-Effectiveness

The initial cost of AI-based DR screening systems is another significant barrier. Although AI technologies can reduce long-term healthcare costs by enabling early detection and intervention, the upfront investment is substantial. This includes the cost of purchasing high-quality retinal imaging equipment, software licenses, and the necessary IT infrastructure to support AI applications. 

An economic modeling study in Singapore suggested that the incorporation of an AI algorithm as an assistive tool in a large-scale DR screening program is associated with significant cost savings [152]. Using a decision tree model, the study compared traditional human assessment with two deep learning approaches: a semi-automated DL model as a triage filter before secondary human assessment, and a fully automated DL model without human assessment. From the health system perspective, the semi-automated screening model was the least expensive of the three models, costing USD 62 per patient per year, generating 19.5% in cost savings compared to the human assessment model.

Other economic modeling studies have reported that the implementation of automated systems for DR screening generates cost savings between 12% and 23.3% in the UK and US [153,154]. In a study analyzing the cost-effectiveness of AI for DR detection in rural China, where patients have limited access to skilled ophthalmologists, AI screening was determined to be cost-effective.

#### 6.4.3. Infrastructure and Access

One of the foremost challenges in implementing AI for DR screening is the lack of necessary infrastructure, particularly in low-resource settings. High-quality retinal cameras, which are essential for capturing the detailed images required by AI algorithms, are often expensive and not readily available in these areas. These costs may be prohibitive, making it challenging to justify the investment despite the potential long-term benefits. Additionally, some low-resource settings lack reliable internet access, which is critical for cloud-based AI systems that require data transmission to central servers for analysis. In many low-resource settings, healthcare disparities already exist, and the introduction of advanced technologies risks widening these gaps if not implemented thoughtfully. Efforts must be made to ensure that AI-based screening programs are accessible to all patients, regardless of socioeconomic status or geographic location. This may involve subsidizing costs, providing mobile screening units, and developing user-friendly technologies that do not require extensive training or infrastructure.

Recent advancements include the integration of AI algorithms with smartphone-based imaging, facilitating reliable DR screening even in remote areas with high sensitivity and specificity (Table 3 and Table 4) [28,155]. Multiple studies, including that by Tomic et al., have validated the effectiveness of hand-held cameras and AI grading systems in DR screening, demonstrating accuracy comparable to traditional clinical examinations [30,156]. However, significant sources of heterogeneity in diagnostic efficacy include variable experience levels of the camera operator, rates of ungradable images due to the inclusion of poor-quality images or media opacity-causing diseases, undilated patients, and overfitting due to small sample sizes. Smart-phone-based imaging presents transformative potential for DR screening, especially in resource-limited settings, by enabling efficient and accurate early detection and facilitating timely referral for treatment.

## 7. Hematology

Research suggests that elevated levels of specific blood markers and cytokines may signal the early stages of DR. Hyperglycemia and the metabolic disruptions it causes trigger a cascade of harmful effects on the retina’s neurovascular structure. These effects impact not only blood vessels but also the optic nerve, glial cells, and immune cells.

### 7.1. Neutrophil–Lymphocyte Ratio

The neutrophil–lymphocyte ratio (NLR), a new inflammatory marker, reflects both innate and adaptive immune responses. The NLR suggests abnormal immune system activity, potentially indicating subclinical inflammation. This type of low-grade inflammation is a common feature of chronic diseases, and individuals with diabetes often exhibit higher NLR levels [157]. Moreover, it has been reported that NLR could be used as a biomarker to predict the incidence of DR in the Scottish population [158]. More recently, El-Tawab et al. reported that NLR showed promise as a reliable marker for detecting DR even before symptoms appear. Studies have shown good sensitivity (89.29%) and specificity (84.37%) for NLR with a cut-off point ≥ 1.97 in identifying pre-clinical DR [114]. Therefore, routine NLR measurement in patients with type 2 diabetes could be beneficial. This simple test could help select individuals suspected of having pre-clinical DR, allowing for earlier intervention.

### 7.2. Neutrophil Extracellular Traps (NETs)

Beyond their role in fighting infection, neutrophil extracellular traps (NETs), webs of chromatin fibers and antimicrobial peptides released by neutrophils, have recently been implicated in the development of various non-infectious diseases, including diabetic retinopathy [159]. Neutrophil elastase (NE), a key component of NETs, has been linked to the early stages of DR. Research suggests it plays a role in capillary degeneration, retinal oxidative stress, and inflammation—all factors contributing to the development of diabetic retinopathy [160]. Like NLR, further research is warranted to identify the cutoff value of prompts for predicting DR occurrence in the early stage.

### 7.3. Ethanolamine

Ethanolamine, also known as monoethanolamine, aminoethanol, or glycinol, exists as free ethanolamine in normal human bodily fluids, such as blood, and its average concentration is about 1.6 μmol/L in the serum of individuals more than 18 years old as revealed in the Human Metabolome Database (https://hmdb.ca/metabolites/HMDB0000149, accessed on 25 March 2024). More recently, ethanolamine was found to significantly lower the serum of DR in individuals with glucose-well-controlled diabetes mellitus (GW-DR) compared to matched control patients and was identified and validated as a potential biomarker for new-onset DR in diabetic patients with well-controlled glycemia. The diagnostic accuracy of ethanolamine for GW-DR ranged from 83.6% to 100% in the discovery cohort and 83.2% to 96.0% in the validation cohort, demonstrating significant improvement over hemoglobin A1c (HbA1c) [161]. Similarly, in two large Asian cohorts, including 464 diabetic patients with various DR stages and 1405 diabetic patients without DR, lower ethanolamine levels in urinary metabolites were associated with DR outcome [162]. Future studies are warranted to investigate the combined analysis of serum and urinary ethanolamine levels to establish reference ranges for the early diagnosis of DR.

## 8. Conclusions and Future Directions

Currently, there is increasing acceptance of the concept that DR is a neurovascular disorder, characterized by dysfunction in the neurovascular unit (NVU) consisting of neurons, glial cells, and vascular cells [163,164,165,166]. The intricate interaction between neurons and glial cells in neurovascular coupling plays a critical role in preserving the normal homeostatic function of the NVU. Consequently, neurodegeneration and glial activation are recognized as primary events in the development of DR, frequently manifesting before the emergence of evident microangiopathy. This phenomenon has been consistently observed in both experimental models of DR and in the retinas of diabetic donors [167]. The pathogenesis of DR lies in the intricate interplay between components of the NVU, resulting in a complex interdependence of structural and functional changes.

An essential question arises: Which of these changes occur earlier and can be detected using current diagnostic methods with high diagnostic power for discriminating eyes of patients with DM who exhibit no to mild DR? This review has delved into this topic in detail, aiming to elucidate the early structural and functional alterations in DR. A preprint study proposed that by employing histological phenotyping and quantitative analysis of postmortem retina from diabetic donors without clinical DR, the observed disparity between localized capillary dropout and widespread neural loss within the inner nuclear layer (INL) suggests that microvascular loss might not directly lead to neurodegeneration during the early stages of DR. This indicated that diabetes could independently impact these two indicators [168]. Both our review and this article suggest that combining structural and functional examinations may represent the optimal strategy for enhancing early detection of DR. Healthcare institutions can implement both structural and functional examinations for patients at high risk of developing DR, tailoring the approach according to their respective capabilities and resources.

In the future, there is a pressing need for more effective, accessible, and reliable methods to facilitate early detection and longitudinal monitoring of DR progression. Of note, the ethical considerations and regulatory challenges associated with the clinical translation of novel approaches for early detection of DR should be kept in mind. Ethically, upholding patient safety through meticulous pre-clinical and clinical trials to rigorously evaluate the safety profile, potential risks, and adverse effects of novel approaches is of paramount importance. Moreover, equitable access to innovative treatments, transcending barriers of affordability and geographic constraints, must be addressed to prevent disparities in care. Consideration must also be given to the inclusion and protection of vulnerable populations, such as children and pregnant women. On the regulatory front, navigating the intricate approval processes, which demand comprehensive data demonstrating safety, efficacy, and quality, can be a protracted and complex endeavor.

Recently, a functional OCTA (fOCTA) system was developed and utilized in diabetic mice [169]. The results revealed that, in normal mice, retinal capillaries displayed a noticeable hyperemic response to flicker light stimulation, whereas diabetic mice exhibited a significant loss of functional hyperemia at an early stage of DR, despite showing few overt signs of retinopathy. If this non-invasive technology can be applied to patients in the future, retinal capillary functional hyperemia may hold strong potential to serve as more sensitive vascular biomarkers of early DR. New approaches have also been explored in animal models of DR for in vivo studies. Notably, Zhang et al. [170] developed an innovative adhesive fluorescent nanoprobe crafted from biodegradable materials. This nanoprobe was designed to detect alterations in VEGFR-2 expression within the retinal microcirculation, offering a non-invasive means of diagnosis. Through specific binding to retinal microvascular endothelial VEGFR-2, the nanoprobe effectively differentiated diabetic animals from their healthy counterparts, showcasing its potential for early detection in DR. Furthermore, novel imaging techniques should be developed and improved for early detection of DR, such as photoacoustic microscopy [171,172,173,174,175,176,177,178]. 

Additionally, efforts should be made to increase the accessibility and convenience of these technologies in real-world settings, particularly in resource-limited settings where the burden of DR is often highest [179]. This may involve the improvement in cost-effective, portable, and user-friendly imaging devices, as well as the integration of telemedicine and AI-assisted analysis for remote screening and monitoring. Longitudinal studies incorporating multimodal imaging and comprehensive clinical data should be undertaken to deepen our understanding of the complex pathophysiology of DR and enhance our ability to detect and monitor DR progression at an early stage. Such research endeavors may pave the way for personalized risk stratification and tailored therapeutic interventions, ultimately improving visual outcomes and vision-related quality of life for individuals with diabetes.

## Figures and Tables

**Figure 1 biomedicines-12-01405-f001:**
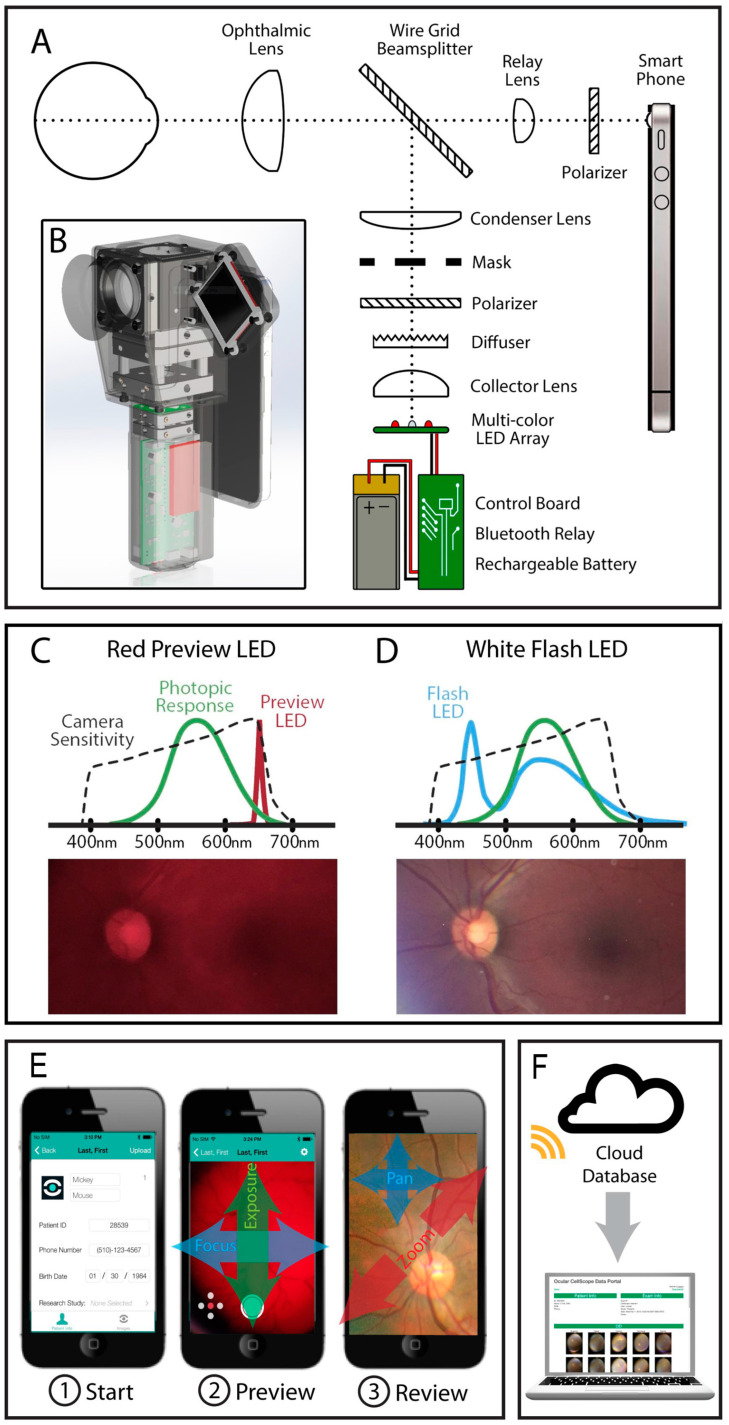
CellScope Retina schematic and workflow. (**A**) Schematic of the optical system. Light from LEDs is directed through a mask and forms an annulus that passes through the peripheral cornea, focusing through the pupil. In propagating through the eye, the light becomes defocused, providing even illumination at the retina. Polarization filters minimize unwanted reflections from anterior ocular surfaces, enabling the smartphone to capture a clear image of the retina. (**B**) The compact optical system and custom-control electronics fit inside a handheld enclosure. (**C**) Red LED illumination of 655 nm peak emission is used for focusing on the retina, which is within the spectral range of the iPhone camera but outside the peak photopic response of the eye. (**D**) A white LED with a broad emission spectrum is flashed for recording images of the retina. LED spectra in (**C**,**D**) are from respective datasheets; photopic response is from the CIE 1931 standard [31]; phone response is approximate for a CMOS phone sensor. (**E**) Smartphone user interface enables (1) patient data capture, (2) preview during focus/alignment with swipe gestures adjusting camera settings, and (3) exam data review with pinch and swipe gestures for browsing stitched image montages. (**F**) Photos can be uploaded directly from the smartphone to a cloud database allowing remote diagnosis with a web interface. Reproduced with permission from Reference [24].

**Figure 2 biomedicines-12-01405-f002:**
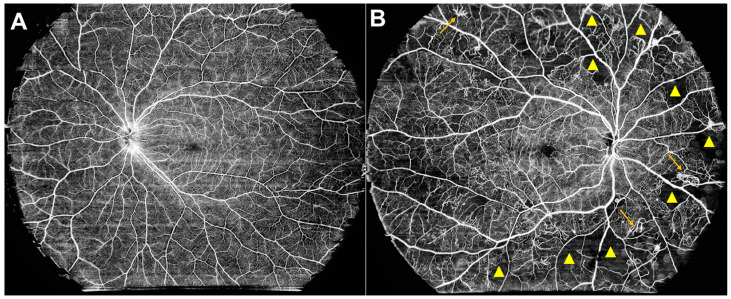
Ultra-wide-field swept-source optical coherence tomography angiography (SS-OCTA) imaging systems in a normal eye (**A**) and in a case of proliferative diabetic retinopathy (**B**). The triangles highlight areas of non-perfusion, while the arrows indicate the presence of retinal neovascularization. (Images courtesy of Dr. Jialiang Duan, The Second Hospital of Hebei Medical University, Shijiazhuang, Hebei, China).

**Figure 3 biomedicines-12-01405-f003:**
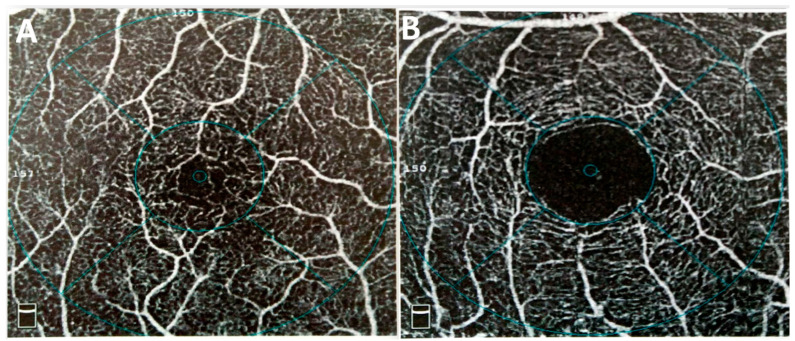
High inter-person variability among normal individuals in the foveal avascular zone (FAZ). (**A**) A woman with a spherical equivalent of +0.75 diopters and an axial length of 24.01 mm exhibited a FAZ area of 0.070 mm^2^. (**B**) A man with a spherical equivalent of −3.50 diopters and an axial length of 23.97 mm showcased a notably larger FAZ area, measuring 0.671 mm^2^.

**Table 1 biomedicines-12-01405-t001:** A summary table comparing the advantages, disadvantages, and key findings in early DR for a variety of retinal imaging modalities.

Imaging Modality	Advantages	Disadvantages	Key Findings in Early DR
Conventional color fundus photography	The standard for evaluating and screening DR. Widely available, cost-effective, and easy to perform. Recent smartphone-based fundus imaging systems are low-cost and portable.	Unable to detect subclinical retinal alterations that occur prior to the appearance of microaneurysms. Limited field of view (between 20° and 50°), may miss peripheral lesions and requires pupil dilation.	Microaneurysms Retinal hemorrhages Macular edema Retinal venular widening Arteriolar tortuosity Venular tortuosity Decreased fractal dimension
Ultra-wide-field fundus photography	Extensive view of periphery (up to 200° in a single image), allowing for assessment of approximately 80% of retinal surface area. Enables detection of milder and earlier forms of DR affecting the periphery. Some systems offer true-color retinal images and artifact mitigation, enabling higher accuracy in detecting microaneurysms and retinal hemorrhages.	Higher cost and less readily available in clinical settings. May require specialized training to operate and interpret findings, particularly in peripheral regions which can appear distorted.	Peripheral retinal lesions and ischemia, in addition to the above findings seen in conventional fundus photography.
Fundus fluorescein angiography (FFA)	Detailed visualization of blood vessel abnormalities and leakage.	Invasive, potential allergic reactions to dye, and requires skilled interpretation.	Microaneurysms Neovascularization (NV) Areas of non-perfusion (NP) Increased foveal avascular zone (FAZ) area
Optical coherence tomography (OCT)	High-resolution images of retinal layers, non-invasive, and quick. Provides quantitative measurements of retinal thickness and volume. Detects subtle retinal changes at early stages of the disease.	Higher cost and less ubiquitous than color fundus cameras. Limited field of view and may miss peripheral pathology. May require specialized training to interpret findings.	Faster rate of GC-IPL thinning Decreased pRNFL thickness Increased inner retinal reflectivity Decreased choroidal vascular index (CVI) Decreased choroidal thickness (ChT) Macular edema Intraretinal fluid or retinal thickening due to fluid accumulation
Optical coherence tomography angiography (OCTA)	Non-invasive, dye-free, and quick. High-resolution, depth-resolved angiographic images, including better visualization of capillary microvasculature compared to FFA. Provides a broader perspective of the fundus and quantitative features for assessing disease severity.	Higher cost and less ubiquitous. Artifacts may affect image quality. Barriers to widespread adoption include scan quality and lack of standardization of quantitative metrics between different commercial OCTA machines.	Decreased perfusion density of the deep capillary complex (DCC) Decreased mid-large choroidal vessel thickness and vessel density Foveal avascular zone (increased area, shape irregularity) Areas of non-perfusion (NP) Microaneurysms
Ultra-wide-field swept-source OCTA (SS-OCTA)	Extensive field of view, in addition to traditional OCTA advantages.	High cost, complex technology, and limited availability.	Early changes in vascular density of the superficial and deep vascular complexes in the peripheral retina, in addition to the above findings for OCTA.

**Table 2 biomedicines-12-01405-t002:** Ophthalmology AI devices approved for DR screening in the US and EU (class II, CE mark).

Model	Approval Year	Target Disease	Company	Markets Available
RetmarkerDR	2010	DR, AMD	Retmarker SA, Taveiro, Portugal	EU
Automated Retinal Disease Assessment (ARDA)	2016	Referable DR	Google LLC, Mountain View, CA, USA	EU
LumineticsCore (previously IDx-DR)	2018	Referable DR, DME	Digital Diagnostics Inc., Coralville, IA, USA	US, EU
OphtAI	2019	Referable DR, DME, AMD, glaucoma	Evolucare/ADCIS, Villers-Bretonneux, France	EU
SELENA+	2019, 2020	Referable DR, AMD, glaucoma	EyRIS Pte Ltd, Singapore	EU, Singapore
EyeArt	2015, 2020	Referable DR, AMD, glaucoma	Eyenuk, Inc., Woodland Hills, CA, USA	US, EU
VUNO Med-Fundus AI	2020	Referable DR, AMD, glaucoma	VUNO Inc., Seoul, Korea	EU, Korea, Singapore
RetCAD	2022	Referable DR, AMD	Thirona Retina BV, Nijmegen, Netherlands	EU
Medios AI	2023	Referable DR, glaucoma	Remidio Innovative Solutions Pvt Ltd., Karnataka, India	EU
AEYE-DS	2024	Referable DR	AEYE Health, Inc., New York, NY, USA	US

**Table 3 biomedicines-12-01405-t003:** Patient level sensitivity and specificity of all three graders as derived using a standard 2 × 2 matrix and Wilson confidence intervals. Reproduced with permission from Reference [28].

Grader Diagnosis	RWDR	No RWDR	Sensitivity (95% CI)	Specificity (95% CI)
Grader 1				
RWDR	52	8	96.3% (86.2, 99.4)	42.9% (18.8, 70.4)
No RWDR	2	6		
Grader 2				
RWDR	49	7	92.5% (80.9, 97.6)	50.0% (24.0, 76.0)
No RWDR	4	7		
EyeArt^®^ AI eye screening system				
RWDR	47	3	87.0% (74.5, 94.2)	78.6% (44.8, 94.3)
No RWDR	7	11		

RWDR, referral-warranted diabetic retinopathy; AI, artificial intelligence.

**Table 4 biomedicines-12-01405-t004:** Eye level sensitivity and specificity of all three graders as derived using a GEE logistic regression with an exchangeable working correlation matrix. Reproduced with permission from Reference [28].

Grader Diagnosis	RWDR	No RWDR	Sensitivity (95% CI)	Specificity (95% CI)
Grader 1				
RWDR	83	14	94.0% (85.5, 97.7)	52.2% (33.4, 70.5)
No RWDR	5	17		
Grader 2				
RWDR	77	10	89.5% (79.3, 95.0)	66.9% (48.8, 81.1)
No RWDR	9	21		
EyeArt^®^ AI eye screening system				
RWDR	69	8	77.8% (67.3, 85.7)	71.5% (48.7, 86.9)
No RWDR	19	23		

RWDR, referral-warranted diabetic retinopathy; AI, artificial intelligence.

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
