# Peer review of "Advances in Structural and Functional Retinal Imaging and Biomarkers for Early Detection of Diabetic Retinopathy"

_biomedicines, 2024, doi:10.3390/biomedicines12071405_

Round 1

Reviewer 1 Report

Comments and Suggestions for Authors

The review paper, “Advances in Structural and Functional Retinal Imaging and Biomarkers for Early Detection of Diabetic Retinopathy,” evaluates advances in retinal imaging and biomarkers for early diabetic retinopathy (DR) detection. It explores multimodal imaging, functional tests, and AI applications and emphasizes their integration for comprehensive early DR assessment, which is highly relevant.

Overall, the manuscript is well put together and thoroughly reviews structural and functional imaging modalities relevant to DR, including emerging techniques such as OCTA and various functional tests like multifocal electroretinography and visual evoked potentials. The manuscript also effectively discusses -supported by recent studies, the potential of AI/ML in enhancing DR detection and diagnosis. The discussion on integrating AI for automating lesion detection and risk stratification is particularly relevant. The authors have done a commendable job integrating insights from multiple specialties, enriching the review’s scope.

However, some methodological gaps need to be addressed. The paper could better detail the methodologies of the studies reviewed to clarify how the conclusions were drawn, particularly the specifics of AI model training and validation. There is also a limited discussion on the practical implications. While advanced technologies are well-discussed, the review lacks in-depth consideration of their practical, real-world applicability and accessibility, especially in low resource settings.

I also put forward that there might also be a potential bias in literature selection! The selection of sources primarily focuses on highly technical and positive outcomes without much critique or discussion of contradictory findings or failures, which may present a biased view of the analysis. Please incorporate any negative findings (if any!) for a thorough and robust discussion of the DR AI applications. This would bring a more balanced view of the literature and enhance the paper’s objectivity and depth.

There is generally a considerable implementation gap in AI, especially in clinical settings. These need to be highlighted. Expanding the discussion to include limitations, potential risks, and the cost-effectiveness of these advanced imaging technologies would provide a more balanced view and assist in clinical decision-making. Perhaps a table would be beneficial here.

Some sections of the paper could benefit from tighter editing to avoid redundancy and improve clarity. I recommend this paper be accepted with these revisions.

Comments on the Quality of English Language

The english is of good quality and requires only minor edits.

Author Response

Response Letter to Reviewer 1

General Reviewer 1 Comment: Overall, the manuscript is well put together and thoroughly reviews structural and functional imaging modalities relevant to DR, including emerging techniques such as OCTA and various functional tests like multifocal electroretinography and visual evoked potentials. The manuscript also effectively discusses, supported by recent studies, the potential of AI/ML in enhancing DR detection and diagnosis. The authors have done a commendable job integrating insights from multiple specialties, enriching the review’s scope. However, some methodological gaps need to be addressed.

Our Response: We appreciate the reviewer for careful reading, constructive feedback, and favorable comments. Our responses to the comments and revision to the manuscript are listed below. The revised manuscript is enclosed. The updated manuscript has undergone extensive revision to include greater discussion on the methodology of various studies implementing AI models, their diagnostic efficacy, and practical implementation challenges.

Specific Reviewer 1 Comments

Specific Comment 1: The paper could better detail the methodologies of the studies reviewed to clarify how the conclusions were drawn, particularly the specifics of AI model training and validation. There is also a limited discussion on the practical implications. While advanced technologies are well-discussed, the review lacks in-depth consideration of their practical, real-world applicability and accessibility, especially in low resource settings.

Our Response: Thank you for this feedback. We have significantly rewritten section 6: “Applications of artificial intelligence in DR detection” (page 14, line 597-784) to provide greater detail on study methodology, model performance, implications, and accessibility by highlighting milestones and market approvals for DR screening algorithms and further details on AI.

Specific Comment 2: I also put forward that there might also be a potential bias in literature selection! The selection of sources primarily focuses on highly technical and positive outcomes without much critique or discussion of contradictory findings or failures, which may present a biased view of the analysis. Please incorporate any negative findings (if any!) for a thorough and robust discussion of the DR AI applications. This would bring a more balanced view of the literature and enhance the paper’s objectivity and depth.

Our Response: Thank you for your thoughtful assessment of potential bias in the literature selection. We appreciate your scrutiny, and we acknowledge the need for providing a balanced discussion on DR AI applications. Where we have found contradictory evidence or negative results, we have made a point to cite these studies (examples include section 2.3.1 Retinal vascular caliber, page 6, line 192; section 2.3.2 Tortuosity of branch retinal artery, page 6, lines 208-214; section 2.3.3 Fractal dimension, page 7, lines 232-233, etc). In section 6 on artificial intelligence, we acknowledge the discussion of opposing or negative findings is limited. Despite our efforts to provide thorough and balanced literature selection, the lack of negative findings likely reflects a degree of publication bias since often negative results are never published in literature. However, to provide a more balanced view, we have included explicit discussion of multiple meta-analyses and discussed sources of heterogeneity in results, although generally positive (page 14, lines 603-609; page 18 lines 776-782).

Specific Comment 3: There is generally a considerable implementation gap in AI, especially in clinical settings. These need to be highlighted. Expanding the discussion to include limitations, potential risks, and the cost-effectiveness of these advanced imaging technologies would provide a more balanced view and assist in clinical decision-making. Perhaps a table would be beneficial here.

Our Response: Thank you for your insight. Based on this feedback, we have included two new subsections, titled “6.3 Potential risks and limitations” (page 17, lines 701-721) and “6.4 Practical implementation challenges” (page 17, lines 722-784) dedicated to discussing these important topics. We have also added Table 2 demonstrating the AI technologies approved in the US and EU.

Table 2: Ophthalmology AI devices approved for DR screening in the US and EU (class II, CE mark).

Model

Approval Year

Target Disease

Company

Market Available

LumineticsCore

(previously IDx-DR)

2018

Referable DR, DME

Digital

Diagnostics

US

EyeArt

2020

Referable DR, AMD, glaucoma

Eyenuk

US

AEYE-DS

2024

Referable DR

AEYE Health

US

RetmarkerDR

2010

DR, AMD

Retmarker

EU

Automated Retinal Disease Assessment (ARDA)

2016

Referable DR

Verily Life Sciences LLC

EU

OphtAI

2019

Referable DR, DME, AMD, glaucoma

Evolucare

EU

SELENA+

2019, 2020

Referable DR, AMD, glaucoma

EyRIS

EU, Singapore

VUNO Med-Fundus AI

2020

Referable DR, AMD, glaucoma

VUNO Inc.

EU, Korea, Singapore

RetCAD

2022

Referable DR, AMD

Thirona

EU

Medios AI

2023

Referable DR, glaucoma

Remidio

EU

Specific Comment 4: Some sections of the paper could benefit from tighter editing to avoid redundancy and improve clarity. I recommend this paper be accepted with these revisions.

Our Response: We really appreciate your thorough feedback and have revised the manuscript throughout for redundancy and clarity.

Reviewer 2 Report

Comments and Suggestions for Authors

Advances in Structural and Functional Retinal Imaging and Biomarkers for Early Detection of Diabetic Retinopathy

This study reviewed current update of diabetic retinopathy, esp diagnistic tools. It is well-recornized review. However, some comments are recommended for better quality of article.

Major comments:

1.     Although several image tools were presented, there are only explanations and no appropriate pictures, so I think it is inadequate in providing information to readers as a review paper. Wide field image for DR suggesting DR. in addition, health retinal image should be provided. Tortuosity image, required drawing. Fractal dimension image, which can not be understandable only text.

2.     2. OCT and OCTA images for DR, should be provided for concise article. Ultrawide field SS-OCTA image should be provide.

Comments on the Quality of English Language

Consisely editing is required for better understanding. It is difficult to read.

Author Response

Response Letter to Reviewer 2

General Reviewer 2 Comment: This study reviewed current update of diabetic retinopathy, esp diagnistic tools. It is well-recognized review. However, some comments are recommended for better quality of article.

Our Response: Thank you for taking the time to provide your constructive feedback and favorable comments. We have revised to improve the clarity of the text and inserted additional figures that summarize key points and provide accompanying visuals.

Specific Comment 1: Although several image tools were presented, there are only explanations and no appropriate pictures, so I think it is inadequate in providing information to readers as a review paper. Wide field image for DR suggesting DR. in addition, health retinal image should be provided. Tortuosity image, required drawing. Fractal dimension image, which cannot be understandable only text.

Our Response: Thank you for your suggestion. Since both tortuosity and fractal dimension of retinal vessels are quantitative variables calculated by software, they cannot be visually demonstrated in images. We have updated the manuscript to add additional figures as recommended, including ultrawide field swept-source optical coherence tomography angiography (SS-OCTA) images of a normal eye and a case of proliferative diabetic retinopathy (Figure 2 in the revised manuscript). The updated manuscript increased the number of Figures and Tables from 4 to 7.

Specific Comment 2: OCT and OCTA images for DR, should be provided for concise article. Ultrawide field SS-OCTA image should be provide.

Our Response: Thank you for your suggestion. We have added Figure 2 in the revised manuscript with ultrawide field SS-OCTA images.

Reviewer 3 Report

Comments and Suggestions for Authors

This manuscript provides a thorough exploration of the latest developments in retinal imaging techniques and biomarkers, aimed at improving the early detection of diabetic retinopathy (DR). It covers a wide range of imaging methods and potential biomarkers that show promise in enhancing our ability to identify DR in its early stages. I believe that this manuscript has the potential to significantly advance the field of ophthalmology and contribute valuable insights to the study of diabetic retinopathy. However, there are some areas where the authors could make improvements to further strengthen the manuscript's impact and credibility.

1.     Would the authors consider incorporating brief paragraphs addressing the clinical progression and molecular components of diabetic retinopathy?

2.  Would the authors kindly consider including the search strategy employed in this review paper in the last sentence of the Introduction? Presenting this information succinctly would greatly enhance the transparency of the methodology for readers.

3.     Would it be possible for the authors to incorporate a discussion on both the pros and cons of each novel approach? This addition would offer readers a comprehensive understanding of the landscape of available modalities.

4.   The authors could consider incorporating a table summarizing human trials or approved applications related to the discussed technologies. Such a table would enhance readability and provide readers with quick access to relevant information, thereby increasing the informative value of the study.

5.    Authors are recommended to make a summary table of different retinal imagining modalities in diabetic retinopathy with advantages, disadvantages, and key findings in DR.

6. Regarding the proposed combination of structural and functional examinations for early detection of DR, could the authors provide more insights into how these two types of examinations complement each other? How might they synergize in improving diagnostic accuracy?

7.     Would the authors kindly comment on the ethical considerations and regulatory challenges associated with the clinical translation of novel approaches for Diabetic Retinopathy? Authors are recommended to explain this part in Section 8. Your insights into this aspect would greatly enrich the readers’ understanding of the practical implications and potential hurdles in bringing innovative treatments to patients.

8.     Considering the potential of novel imaging techniques like photoacoustic microscopy to enhance early detection of diabetic retinopathy, could the authors please kindly share any potential limitations or considerations that must be addressed to ensure their effective utilization?

Comments on the Quality of English Language

The quality of English language was fine. 

Author Response

Response Letter to Reviewer 3

Specific Comment 1: Would the authors consider incorporating brief paragraphs addressing the clinical progression and molecular components of diabetic retinopathy?

Our Response: Thank you for your suggestion. While the clinical progression and molecular components of diabetic retinopathy are not the primary focus of this review article, we have updated the introduction to include some brief information on this. In the first paragraph, we have cited some of the latest review articles, which can provide readers with more detailed information regarding progress of diabetic retinopathy.

Specific Comment 2: Would the authors kindly consider including the search strategy employed in this review paper in the last sentence of the Introduction? Presenting this information succinctly would greatly enhance the transparency of the methodology for readers.

Our Response: Thank you for your suggestion. We have added the search strategy in the last sentence of the Introduction in the revised manuscript (page 2, lines 96-102).

Specific Comment 3: Would it be possible for the authors to incorporate a discussion on both the pros and cons of each novel approach? This addition would offer readers a comprehensive understanding of the landscape of available modalities. Authors are recommended to make a summary table of different retinal imaging modalities in diabetic retinopathy with advantages, disadvantages, and key findings in DR.

Our Response: Thank you for your feedback. We have added in the revised manuscript a summary Table 1, of different retinal imaging modalities with the advantages, disadvantages, and key findings in early diabetic retinopathy (DR) as requested. This table aids in highlighting the key pros and cons embedded in the manuscript as suggested.

Specific Comment 4: The authors could consider incorporating a table summarizing human trials or approved applications related to the discussed technologies. Such a table would enhance readability and provide readers with quick access to relevant information, thereby increasing the informative value of the study.

Our Response: We appreciate your suggestion. We have included in the updated manuscript summary Table 2, to highlight the approved AI applications for DR screening.

Specific Comment 5: Regarding the proposed combination of structural and functional examinations for early detection of DR, could the authors provide more insights into how these two types of examinations complement each other? How might they synergize in improving diagnostic accuracy?

Our Response: Thank you for your query. Prior studies have effectively combined structural and functional exams to elucidate the chronology and pathophysiologic relationship between microvascular alterations and neurodegenerative processes in DR (page 13, lines 549-572). Given the growing body of evidence that neurodegenerative processes precede microvascular complications, functional testing provides another dimension of information that may present as early, if not earlier than many structural imaging biomarkers. Therefore, combining structural with functional data may enhance the performance of AI algorithms in earlier detection of DR, prior to clinical manifestations. Since previous studies did not provide a comparative analysis of the accuracy of different examination methods, our review article is unable to provide a more thorough discussion on this matter. In the first two paragraphs of Section 8, we discuss why and how to utilize structural and functional examinations for early detection of DR. As we have concluded, combining structural and functional examinations may represent the optimal strategy for enhancing the early detection of DR in its initial stages (page 20, lines 861-865). We recommend that healthcare institutions implement both structural and functional examinations for patients at high risk of developing DR, tailoring the approach according to their respective capabilities and resources.

Specific Comment 6: Would the authors kindly comment on the ethical considerations and regulatory challenges associated with the clinical translation of novel approaches for Diabetic Retinopathy? Authors are recommended to explain this part in Section 8. Your insights into this aspect would greatly enrich the readers’ understanding of the practical implications and potential hurdles in bringing innovative treatments to patients.

Our Response: Thank you for your suggestion. We have added this part in the revised manuscript (page 21, lines 868-878).

Specific Comment 7: Considering the potential of novel imaging techniques like photoacoustic microscopy to enhance early detection of diabetic retinopathy, could the authors please kindly share any potential limitations or considerations that must be addressed to ensure their effective utilization?

Our Response: Photoacoustic microscopy (PAM), while very promising in preclinical animal models of diabetic retinopathy like the DL-AAA model, has yet to be formally implemented in clinical settings with no FDA approved devices for ocular imaging. However, it is anticipated that the future integration of PAM and nanoparticles may facilitate the early detection of microangiopathy and molecular changes of individuals with diabetic retinopathy. We include a sentence on PAM along with giving key references on the technology.

Round 2

Reviewer 1 Report

Comments and Suggestions for Authors

I thank the authors for considering my input and making the necessary changes. I am satisfied that the manuscript has undergone significant changes through the peer review process and has sufficiently improved to recommend its publication. I wish the authors good luck in their future scientific endeavors.

Reviewer 2 Report

Comments and Suggestions for Authors

Dear authors

After the revision, MS became more improved and concise.

Comments on the Quality of English Language

English is good.

Reviewer 3 Report

Comments and Suggestions for Authors

The authors did their best to revise the manuscript as the reviewer's comments and answer the reviewers' questions.